# Can China's industrial policies enhance the green competitiveness of the manufacturing industry?

Qing Zhao ⬮*◉, Chih-Hung Yuan ⬮*◉

School of Economics and Commerce, University of Electronic Science and Technology of China, Zhongshan Institute, Zhongshan, China

◉ These authors contributed equally to this work.
* zhaoguoqing1977@163.com (QZ); ialexyuan@gmail.com (C-HY)

## Abstract

This study innovatively uses local government regulations related to manufacturing to quantitatively identify industrial policies. The degree of influence and functional mechanism of China's industrial policies on the green competitiveness of the manufacturing industry are empirically examined using the provincial and regional panel data. Additionally, the synergistic complementary effect between industrial policy power and market forces and the fiscal decentralization's role in influencing industrial policies are investigated. The results reveal that the promulgation and implementation of industrial policies have significantly promoted the green competitiveness of the manufacturing industry. Regarding functional mechanisms, environmental governance has played a positive role in promoting the green competitiveness of the manufacturing industry supported by industrial policies, resource allocation, and innovation incentives. Meanwhile, industrial policies on green competitiveness in manufacturing depend on marketization and fiscal decentralization in local governments. The above findings demonstrate that the local governments in China, a developing economy, can play the role of development-oriented governments. Based on conforming to market deepening and system optimization, they can formulate and implement industrial policies in a rational manner and achieve green development and upgrade the manufacturing industry.

## Introduction

China's manufacturing industry has been developing at an astonishing rate since the reform and opening up. From the total output perspective, China has surpassed the United States to become the largest manufacturing nation, established as the world's most stable, complete industrial system, and has played an irreplaceable role in the international industrial division of labor. Nevertheless, overlooking the development status of the manufacturing industry, China has not eliminated the restrictions of the conventional high-pollution, high-consumption, and high-emissions development model. Tight resource constraints and prominent ecological issues remain shortcomings that compromise international competitiveness in the

**Data Availability Statement:** The data underlying the results presented in the study are available from the Statistical Yearbooks of China. China Statistical Yearbook can be obtained from the online community platform of China Home of

Economics and Management (https://bbs.pinggu.
org/). Industrial policy data comes from Chinese
laws and regulations database (http://search.
chinalaw.gov.cn/search2.html).

**Funding:** Guangdong Province Philosophy and
Social Science Planning 2019 General Project
"Research on Industrial Policy Transformation to
Promote High Quality Development of
Manufacturing Industry" (Approval number:
GD19CYJ01). The Ministry of Education of
Humanities and Social Science Project
(19YJC630185).

**Competing interests:** The authors have declared
that no competing interests exist.

manufacturing industry. China is a nation that implements many industrial policies. In modern Chinese industry, industrial policies are major tools for the government to guide industrial development. In recent years, industrial policies have become a more important means of "adjusting structure and promoting transition" for China. Can industrial policies enhance the green competitiveness of China's manufacturing industry? How do industrial policies impact green competitiveness in manufacturing? What influence will China's unique institutional environment exert on the implementation of industrial policies? An appropriate understanding of these issues will be beneficial for the government to improve and coordinate relevant industrial policies continuously, promote the high-quality development of the manufacturing industry, and enhance its green competitiveness.

Given the broad application of industrial policies worldwide, the effectiveness of industrial policies has always been the focus of both academia and industry. According to the "ineffectiveness theory" of industrial policies, they can hardly attain the expected effect due to government failures, even descending to rent-seeking tools. Meanwhile, government failures primarily originate from the limited cognitive ability of the government, its pursuit of short-term benefits, and insufficient information and incentive distortion [1,2]. There has been substantial empirical evidence supporting the conclusion that "industrial policies are ineffective." For instance, trade protection policies lower the growth rate of total factor productivity (TFP) [3]; tariff protection fails to significantly improve labor productivity in the manufacturing industry [4]; policies such as export subsidies compromise the competitiveness of the steel industry in various countries [5]; aid between EU countries is significantly negatively correlated with industrial performance [6]; and local governments' industrial policies lead to declined geographic concentration and the degree of specialization of industries [7].

In contrast to the advocate for the market mechanism by ineffectiveness theorists, the effectiveness theorists believe that market failures invalidate the market mechanism, and that industrial policies can improve social welfare by correcting market failures [8,9]. Regarding empirical research, the East Asian Miracle itself is the greatest support for the effectiveness theory of industrial policies. Additionally, according to an investigation of the German industrial agglomeration policy by Falck et al. [10], the innovation rate of supported businesses increased slightly, and the cost for research and development was lowered drastically. Harris et al. [11] examine the effects of Canada's protective tariffs and concluded that the scale, productivity, and innovation capability of protected industries had improved remarkably. Bae and Mah [12] discuss the role of industrial policy in the economic development of Uzbekistan, which took a gradualist approach in transition and continued to record rapid economic growth since the early 2000s. Davis and Renski [13] provide evidence of how various measures of urban industrial activity change following the designation of an industrial preservation policy. This research suggest that industrial preservation policies can be an effective tool to stem the urban industrial land loss in cities facing land-use conversion pressures. Pang et al. [14] find that government subsidies, tax incentives, and government procurement exert a positive synergistic effect on innovation. Agricultural non-point source pollution control and prevention policies have contributed to pollution reduction [15].

Reviewing extant studies on the effectiveness of industrial policies, we find that despite the research achievements by scholars, there remain distinct shortcomings: First, extant studies on industrial policy effectiveness focus primarily on the relationship between industrial policies and labor productivity. However, little research has examined how industrial policies affect green TFP (GTFP). Second, in the existing literature, the institutional environment for successful industrial policies has rarely been analyzed. According to Cimoli et al. [16], a successful industrial policy depends on the combined effects of factors and institutions. Among them, factors can be understood as subsidies to private sectors or investments in human capital. Simultaneously, institutions enable factors such as subsidies and investments to act on

economic growth rather than "entering the pocket of rent-seekers." Hence, studying the institutional foundation, by following Cimoli et al.'s ideas, for the effect of industrial policy implementation is necessary. Finally, in the extant literature, the measures of industrial policies focus on aspects such as tariff subsidies, tax incentives, and innovation subsidies. However, these measures are tools that cannot cover the entire content of industrial policies.

The contribution of this paper, as compared to previous research, is reflected mainly in three aspects: First, from an industrial policy perspective, despite productive discussions on the relationships between industrial policies and productivity [11], corporate innovation [10], and economic growth [17], few have studied the economic effects of industrial policies from the green competitiveness perspective. This study finds that industrial policies greatly influence green competitiveness in manufacturing. Accordingly, an in-depth analysis concerning the specific mechanisms and institutional environment during the implementation of industrial policies is conducted, extending the relevant research on the effects of industrial policy implementation. Second, from the GTFP perspective, previous studies have focused on exploring how environmental regulation impacts GTFP [18–20]. Nevertheless, to cope with the challenges of sustaining economic growth and avoiding environmental disasters, more is required than just improving the resource allocation efficiency by internalizing environmental costs [21]. It does not produce any development path consistent with broader social objectives. Such a process of development concept transition from externality to social objectives cannot be coordinated simply by environmental regulation. Thus, it is necessary to analyze sustainable development from the industrial policy perspective as an ultimate goal within the ecosystem boundaries. This study examines the effects and specific mechanisms of the manufacturing industry by which local industrial policies influence the GTFP, enriching the relevant research on the influencing factors of GTFP, considering industrial policies as the starting point. Third, industrial policies are the sum of various policies proposed by the government for industrial formation and development to attain certain economic and social targets. This study measures industrial policies based on governmental regulatory documents and verifies the rationality and feasibility of such an approach, thereby expanding the perspective of extant research on industrial policies.

## Empirical model and variable description

### Empirical model settings

The following empirical model is developed to verify the impact of governmental industrial policies on green competitiveness in manufacturing:

$$lnGTFP_{it} = C + \beta_1 Policy_{it} + \beta_2 \text{Export}_{it} + \beta_3 Capital_{it} + \beta_4 FDI_{it} + V_i + \varepsilon_{it} \qquad (1)$$

where i denotes the region, and t denotes time. This study uses the panel data of 30 provinces and regions in China during 2004–2016. In the data, $GTFP_{it}$ denotes the green competitiveness level of the manufacturing industry in the t-th year for the i-th province. The measured GTFP is adopted in this study. Policy is a variable that represents the governmental industrial policy; X denotes the control variables, which include the degree of openness (Export, FDI) and human capital (Capital), C is the intercept independent of individuals, β is the parameter under estimation, $V_i$ is the individual effect, and $\varepsilon_{it}$ is the stochastic error term. Table 1 provides descriptive statistics for the main variables.

### Estimation of green competitiveness in the manufacturing industry

Measuring methods. Derivation of GTFP by incorporating factors such as energy consumption and environmental pollution into the TFP computation framework is an important

**Table 1. Descriptive statistics of major variables.**

| Variable | Obs | Mean | Std. Dev. | Min | Max |
|----------|-----|------|-----------|-----|-----|
| GTFP | 390 | 0.509 | 0.609 | -1.839 | 2.028 |
| Policy_acc | 390 | 3.514 | 1.292 | 0 | 5.627 |
| Policy_add | 390 | 1.992 | 1.029 | 0 | 4.110 |
| Capital | 390 | 10.727 | 1.229 | 7.930 | 14.940 |
| FDI | 390 | -2.247 | 1.277 | -5.163 | 1.109 |
| Export | 390 | 4.982 | 0.977 | 3.077 | 7.118 |
| Subs | 390 | 0.193 | 0.193 | 0.013 | 0.824 |
| Inno | 390 | 5.439 | 2.997 | 1.491 | 14.876 |
| Gov | 390 | 6.278 | 4.095 | 0.0713 | 0.071 |
| Market | 390 | 6.319 | 1.7778 | 2.530 | 10.920 |
| FD | 390 | 0.221 | 0.237 | 0.007 | 1.834 |

advancement in productivity research. Initially, scholars extended the conventional Cobb-Douglas production function model, which incorporated the energy loss and pollutant emissions into the production function as input factors together with labor and capital to measure the GTFP. However, dealing with environmental indicators, input factors go against the "concept of material balance [22]." Later, scholars proposed a directional distance function (DDF)-based analysis model of environmental regulatory behavior. With this method, environmental pollution was introduced into the production process as an undesirable "bad" output [23]. The problem is that DDF requires the input or output to change equi-proportionally (radially), and making input- or output-based choices (angular) is also needed during efficiency measurement. To overcome these two shortcomings, Chuang et al. [24], Kumar [25], and Oh et al. [26] constructed the Malmquist-Luenberger (ML) TFP index based on the DDF, where the undesired output was used as the output variable to measure the GTFP that considered environmental factors to overcome these two shortcomings. This study employs the DDF-GML approach to estimate the GTFP of the manufacturing industry, used to measure green competitiveness.

Selection of green competitiveness input and output indices. According to the aforementioned theoretical method, the good output, bad output, and input quantity data across China were needed for 2004–2016. Good output is converted into the actual gross output of the local manufacturing industry by taking 2004 as the base year based on the producer price indices of industrial products in various regions. Industrial $SO_2$ emissions, wastewater, and solid waste discharge in various regions were selected regarding bad output. Meanwhile, input variables include fixed capital stock, number of employees at the year-end, and energy input.

Capital input. The capital stock is estimated based on the value of fixed assets using the perpetual inventory method, where problems in four aspects need to be considered: the choice of current investment indices, capital stock in the base period, depreciation rate, and investment deflation, to estimate the region-wise capital stock of China's manufacturing industry. The specific steps are as follows: (1) Calculation of depreciation rate. In the extant literature, a constant depreciation rate is often used for capital stock estimation, a crude practice. The 2004–2008 *Statistical yearbooks of China's industrial economy* offer region-wise current depreciation and fixed asset costs of manufacturing industries above the designated size for 2004–2007. By utilizing the ratio of depreciation in the current year to the fixed asset cost in the previous year, the corresponding depreciation rate can be calculated. Estimating depreciation rates for 2008–2016 is also necessary. Since the statistical yearbooks offer region-wise complete data on the accumulative depreciation and fixed asset costs for this period, it is possible to infer the

implied depreciation rate based on the intrinsic relationships between variables. Depreciation rate $_t$ = depreciation in the current year $_t$/ cost of fixed assets $_{t-1}$ = (accumulated depreciation $_t$—accumulated depreciation $_{t-1}$)/ cost of fixed assets $_{t-1}$. (2) Calculation of new annual investments in fixed assets investment in the current year $_t$ = cost of fixed assets $_t$—cost of fixed assets $_{t-1}$. Finally, with the region-wise price index of investment in fixed assets for the manufacturing industry, the 2004–2016 manufacturing sector's region-wise full-caliber volumes of investment in the current year are deflated to a comparable price sequence at the price level in 2004. The price index of investment in fixed assets is obtained from the *Statistical Yearbooks of China* over the years. (3) Determination of initial capital stock in 2001. The extant region-wise full-caliber data on the net value of fixed assets for the manufacturing sector in 2004 are further converted into a comparable net fixed asset value in 2004 based on the price index of investment in fixed assets. These data are used as the initial capital stock for 2004. (4) Estimation of capital stock using the perpetual inventory method. Based on the first three steps, the region-wise capital stock of the manufacturing industry is calculated: capital stock $_t$ = comparable full-caliber investment $_t$ + (1—depreciation rate $_t$) × capital stock $_{t-1}$.

Labor input. Labor time is better than the number of laborers to measure the role of labor input, difficult to acquire. Hence, the region-wise annual mean number of all employees from manufacturing firms for 2004–2016 is used to replace labor time.

Energy input. Resources serve as an intermediate input, and the energy input is not considered in the conventional TFP. This study considers the energy input, assumed to be the primary source of "bad" output. The sum of various industrial energy consumptions converted to 10,000 tons of standard coal, published in the regional energy balance sheets of the *China Energy Statistical Yearbooks* over the years, is used as an indicator of industrial energy input. The reference coefficients of various energies converted to standard coal are derived from the *China Energy Statistical Yearbooks*.

The marketization index (MI index) for various regions is calculated using the DDF-GML method in this study. Assuming that the GTFP for the base period of 2004 is 1, the GTFP in 2005 is equal to 1 multiplied by the MI in 2005. Accordingly, the GTFP is calculated for various provinces using MaxDEA 7.0 software.

## Core explanatory variable

In this study, industrial policy is the core explanatory variable. The industrial policy variable (Policy) is defined as Policy_acc (cumulative number of local regulations and documents in the manufacturing industry) and Policy_add (number of newly added local governmental regulations and documents on the manufacturing industry). Industrial policies refer to a series of policies using which the government intervenes in resource allocation and benefits distribution, restricts (compulsory), induces (incentives) corporate behavior, and influences the direction of industrial development. The quantitative assessment of industrial policies is a frontier issue that spans academic and policy communities. To empirically examine the rationality assertion of industrial policies, demanding abstraction and quantification of "industrial policy" are necessary, a policy behavior variable. Some scholars have attempted to perform quantitative analyses. Aghion et al. [27] chose tax incentives and government subsidies in China as indicators for measuring industrial policies, analyzing their impact on the TFP separately from the micro and macro perspectives. However, tax incentives and government subsidies should be classified as fiscal policies rather than industrial policies. This study finds that China's industrial policies rarely appear as laws. Most appear as departmental regulations and local governmental regulatory documents. Hence, measuring industrial policies from the perspective of the number of regulatory documents is a feasible approach.

After collation and manual selection based on the laws and regulations database of China, Policy_acc is derived for each province and city across each year. The specific processing procedure is as follows: First, according to the provisions on the effectiveness hierarchy of laws, the departmental regulations have general binding or guiding effects on all provinces and cities throughout China as a higher-level law, excluded in this study. Second, industry regulations and group rules can only be legally valid for individual industries or members of mass organizations, excluded in this study. Third, in terms of timeliness, there are five states of industrial policy documents: currently valid, revised, corrected, invalid, and partially invalid. The invalid samples are removed from the local government regulations of various years. The revised, corrected, and partially invalid documents are still regarded as the continued implementation of original policies and are not deleted.

## Control variables

Based on the extant literature, we use provincial-level features in the regression analysis as control variables to minimize the omitted variable bias, including export demand (Export), foreign direct investment (FDI), and human capital (Capital). FDI is expressed as the actual use of foreign investment in each region as a percentage of GDP. FDI can be used to verify the "pollution haven hypothesis" and the "green haven hypothesis [28]. According to [29], FDI negatively impacts the productivity of Chinese factories, while Choe [30] claimed that FDI has a positive effect on China's environmental TFP. Thus, it is necessary to examine the impact of FDI on green competitiveness in the manufacturing industry. The data were obtained from statistical yearbooks of various regions over the years. The variable Export is measured by the export volume in each region as a percentage of the GDP. China is the world's largest exporter. Although exports help expand foreign demand and provides funding for R&D to stimulate innovation, the expansion of exports has exacerbated resource consumption and environmental deterioration in China, which may also increase pollution. Thus, it is necessary to examine its impact on the green competitiveness of the manufacturing industry. The data are obtained from the *Statistical Yearbooks* of *China* over the years. Labor markets, with higher human capital, play a significant role in promoting economies of scale and the facilitation of increasing returns to scale. However, regions with higher agglomeration degrees generally have higher levels of labor advancement, producing a positive effect on green economic efficiency. Since the enhancement of human capital often depends on the educational level, this study measures local labor market advancement by using the per capita proportion of years of education.

## Variables of the path and institutional environment

This study selects three paths: resource allocation, innovation, and environmental governance to explore the pathway of industrial policy to green competitiveness of the manufacturing industry. Specifically, resource allocation (Subs) is measured by the proportion of government funds in the sales revenue of the manufacturing industry. Technological innovation (Inno) is measured by the proportion of patents for the invention. The proportion of environmental governance investment measures environment governance (gov) in the sales revenue of the manufacturing industry. Furthermore, two representative Chinese institutional environments, the marketization process and fiscal decentralization, are chosen to study the regulatory function of the institutional environment on industrial policy. This research takes the "MI of various regions in China" compiled by Fan Gang et al. as the proxy variable to measure the marketization process. It uses the proportion of fiscal expenditure within the provincial budget in the gross expenditure of the national budget as the indicator to assess fiscal decentralization.

Table 1 shows the statistical description of the above-explained variables, core explanatory, and control variables.

### Sources of data

The statistical data of the provincial manufacturing industry over 2004–2016 are used for research and analysis. Regarding the data sources, the data of million tons of standard coal used for calculating energy utilization ratio are obtained from *China Energy Statistical Yearbook*; the data of wastewater, waste gas, and waste emissions and SO2 emission are obtained from *China Environment Statistical Yearbook*; and the data for measuring industrial policy are obtained from Database of China Laws and Regulations. The marketization data are obtained from the "MI of Various Regions in China" compiled by Fan Gang et al. Meanwhile, other variables are acquired from *the China Statistical Yearbook*.

## Empirical analysis

### Industrial policies and green competitiveness of manufacturing industry

Table 2 illustrates the regression results of the empirical model (1). Column (1) lists the estimates excluding control variables like FDI, where the regression coefficient of core explanatory variable Policy_acc is 0.249, statistically significant at a 1% level. The control variables Export, FDI, and Capital are added in column (2) to ensure the robustness of the results. No significant changes are noted in the estimates, and the impact coefficient remains significantly positive at a 1% level. In columns (1) and (2) of Table 2, the effective stocks of local government regulations are used as the explanatory variable, verifying that the industrial policies positively affect the GTFP. In Columns (3) and (4) of Table 2, regression analysis is performed by considering Policy_add as the explanatory variable. The impact coefficients of the core explanatory variable Policy_add are always significant regardless of whether the control variables are added or not.

**Table 2. Benchmark regression: Fixed-effects analysis.**

|  | (1) | (2) | (3) | (4) |
|---|---|---|---|---|
|  | GTFP | GTFP | GTFP | GTFP |
| Policy_acc | 0.249*** | 0.157*** |  |  |
|  | (18.39) | (6.06) |  |  |
| Policy_add |  |  | 0.228*** | 0.059*** |
|  |  |  | (11.59) | (2.68) |
| Capital |  | 0.089** |  | 0.186*** |
|  |  | (2.37) |  | (5.34) |
| FDI |  | 0.141** |  | 0.260*** |
|  |  | (2.56) |  | (4.91) |
| Export |  | 0.149*** |  | 0.112** |
|  |  | (3.09) |  | (2.26) |
| _cons | -0.365*** | -1.427*** | 0.054 | -1.575*** |
|  | (-7.32) | (-2.75) | (1.26) | (-2.91) |
| $R^2$ | 0.485 | 0.528 | 0.272 | 0.490 |
| *N* | 390 | 390 | 390 | 390 |

*t* statistics in parentheses

* $p < 0.1$

** $p < 0.05$

*** $p < 0.01$.

The regression coefficient for Policy_add is 0.059 in column (4), showing statistical significance at a 5% level. From the economic significance perspective, for every additional regulatory document issued by the local government, the green competitiveness of the manufacturing industry will be enhanced pronouncedly by 0.059 units.

The coefficients for control variables Export and FDI are all significantly positive. This indicates that China's opening-up policy is beneficial for improving green competitiveness in the manufacturing industry, conforming to the "green haven hypothesis." Additionally, a significantly positive correlation of Capital is found with the manufacturing sector's GTFP, suggesting that the improvement of human capital level greatly influences the regional green competitiveness.

## Robustness test

Table 3 reports the results of the robustness test.

## Introduction of GTFP lag term as the control variable, and GMM-based estimation

Local governments may differ in their capacity and skills to promulgate industrial policies. Regions with higher levels of manufacturing development may introduce more effective and rational industrial policies, affecting the unbiasedness and consistency of model estimation in the presence of reverse causality. Hence, this study employs the system GMM proposed by

**Table 3. Robustness test.**

|  | (1) | (2) | (3) | (4) | (5) | (6) |
|---|---|---|---|---|---|---|
|  | GTFP | GTFP | Addl | Addl | GTFP | GTFP |
| L. GTFP | 0.882*** | 0.955*** |  |  |  |  |
|  | (34.12) | (44.99) |  |  |  |  |
| Policy_acc | 0.096*** |  | 0.323*** |  | 0.166*** |  |
|  | (8.32) |  | (14.81) |  | (4.63) |  |
| Policy_add |  | 0.002*** |  | 0.132*** |  | 0.063** |
|  |  | (3.99) |  | (6.12) |  | (2.04) |
| Capital | 0.137*** | 0.089*** | 0.0704** | 0.266*** | 0.102** | 0.178*** |
|  | (19.77) | (18.56) | (2.21) | (7.78) | (2.04) | (3.66) |
| Export | 0.109*** | 0.073*** | 0.116*** | 0.042 | 0.190*** | 0.147** |
|  | (5.94) | (3.90) | (2.86) | (0.87) | (3.32) | (2.53) |
| lnFDI | 0.077*** | 0.064*** | 0.217*** | 0.453*** | 0.054 | 0.213*** |
|  | (9.30) | (3.44) | (4.67) | (8.71) | (0.67) | (2.83) |
| _cons | 0.887*** | 0.823*** | 2.978*** | 2.652*** | -1.846** | -1.523** |
|  | (11.41) | (5.75) | (6.80) | (4.99) | (-2.56) | (-2.03) |
| $R^2$ |  |  | 0.819 | 0.735 | 0.509 | 0.472 |
| AR(1) | -3.016 (0.002) | -3.105 (0.002) |  |  |  |  |
| AR(2) | 1.1566 (0.24) | 1.400 (0.16) |  |  |  |  |
| sargan | 29.215 (0.93) | 27.273 (0.96) |  |  |  |  |
| N | 360 | 360 | 390 | 390 | 247 | 247 |

*t* statistics in parentheses

* $p < 0.1$

** $p < 0.05$

*** $p < 0.01$.

Blundell & Bond [31] to estimate the aforementioned model and addresses the endogeneity problem existing in the model using the lag term of the explanatory variable as the instrument variable. The system GMM-based estimation requires the second-order sequence to pass the correlation test for the random disturbance term of the difference equation and the Sargan over-identification test on the validity of the instrument variable. According to the test results in Columns (1) and (2) of Table 3, the system GMM-based estimation is effective, and its results maintain extremely high robustness compared to the feasible generalized least squares (FGLS) estimates.

## Replacement of core variable

At the micro-level, the enhancement of competitiveness is reflected in the process in which a firm or an economy moves toward a more profitable capital and technology-intensive economic field. It is a low-to-high transition of value-added activities within the value chain, focusing on improving industrial value-creating ability. Accordingly, Kaplinsky and Readman [32] use product value-added as a measure of competitiveness. This study measures the green competitiveness of the manufacturing industry with the added value per capita (Addl). Columns (3) and (4) of Table 3 show that the regression coefficients remain significantly positive.

## Examination of weak endogenous subsamples

If an endogeneity problem exists between the industrial policies and high-quality manufacturing development, such problems should be more serious in regions with higher GTFP levels in manufacturing. In contrast, if the GTFP level of the manufacturing industry is lower in the samples, the endogeneity problem may be weakened. According to the regression results with weak endogenous subsamples (from the Central and Western regions of China) in Columns (5) and (6) of Table 3, industrial policies still significantly promote the manufacturing industry's GTFP level.

## Search and verification of transmission channels

To further understand why industrial policies can impact the green competitiveness of the manufacturing industry, the following possible transmission channels are considered: resource allocation, innovation, and environmental governance. With these channels, how industrial policies influence the green competitiveness of the manufacturing industry are explored, and Table 4 presents the relevant results.

   Initially, we examine whether industrial policies impact the green competitiveness of the manufacturing industry through resource allocation channels. Local governments offer a series of preferential measures for the green transition of manufacturing, such as expanding the scale of industrial green credit and bonds and subsidizing those who reduce emissions, thereby altering the allocation of factor resources among enterprises to cooperate with the implementation of industrial policies. Cerqua and Pellegrini [33] state that financial subsidies can improve firms' investment level and growth. Slant resources encourage firms to carry out R&D on clean technologies and control pollution emissions, internalizing the positive externalities of energy conservation and emission reduction and motivate corporate enthusiasm to participate in environmental governance effectively, thereby enhancing the green competitiveness of the manufacturing industry. The proportion of government funds in manufacturing sales revenue is chosen as the proxy variable of resource allocation (Subs) to verify this mechanism. Table 4 reports the corresponding empirical regression results. The regression coefficients for the resource allocation variable in Column (1) are significantly positive, indicating that resource allocation has promoted green competitiveness in manufacturing. Meanwhile,

**Table 4. Transmission channels.**

| | (1) | (2) | (3) | (4) | (5) | (6) | (7) | (8) | (9) |
|---|---|---|---|---|---|---|---|---|---|
| | GTFP | Subs | Subs | GTFP | Inno | Inno | GTFP | Gov | Gov |
| Policy_acc | | 0.120*** | | | 0.092*** | | | 0.262*** | |
| | | (6.96) | | | (12.42) | | | (11.16) | |
| Policy_add | | | 0.107*** | | | 0.091*** | | | 0.229*** |
| | | | (4.93) | | | (9.32) | | | (7.38) |
| Subs | 0.269*** | | | | | | | | |
| | (5.14) | | | | | | | | |
| Inno | | | | 0.906*** | | | | | |
| | | | | (9.00) | | | | | |
| Gov | | | | | | | 0.187*** | | |
| | | | | | | | (5.31) | | |
| _cons | 0.348*** | 0.180*** | 0.387*** | 0.308*** | -0.104*** | 0.039* | 1.689*** | 10.830*** | 11.300*** |
| | (9.28) | (2.83) | (8.12) | (10.51) | (-3.78) | (1.84) | (4.08) | (125.35) | (166.01) |
| Control variable | Yes | Yes | Yes | Yes | Yes | Yes | Yes | Yes | Yes |
| Region | Yes | Yes | Yes | Yes | Yes | Yes | Yes | Yes | Yes |
| Year | Yes | Yes | Yes | Yes | Yes | Yes | Yes | Yes | Yes |
| $R^2$ | 0.480 | 0.184 | 0.184 | 0.482 | 0.400 | 0.484 | 0.484 | 0.269 | 0.240 |
| N | 390 | 390 | 390 | 390 | 390 | 390 | 390 | 390 | 390 |

$t$ statistics in parentheses

* $p < 0.1$

** $p < 0.05$

*** $p < 0.01$.

the significantly positive coefficients for the industrial policy variable in Columns (2) and (3) demonstrate that industrial policies have driven the manufacturing industry to acquire more resources.

Next, we examine the technological innovation channels. Michael Porter once highlighted that the sustainable competitive advantage of firms comes primarily from corporate innovation. However, corporate innovation requires substantial R&D investment and features a long R&D cycle, high risk, externality, etc. Due to these features, firms have to face the test of "Death Valley of Innovation," reducing their willingness to invest in innovation, resulting in insufficient investment in innovation [34]. Nonetheless, industrial policies can compensate for lack of innovation and market failure through the government. Innovation subsidies and tax incentives in industrial policy instruments can lower corporate R&D costs to some extent, relieve the financial pressure in R&D investment, and increase the risk of corporate innovation. Government procurement and technology control stimulate firms to increase investment in R&D of new technologies and offer market demand guidance to develop new products and processes. The intellectual property protection and the patent systems provide incentives and protection for corporate R&D and innovation to derive incentives to increase R&D and innovation-driven profits. Therefore, their willingness and ability to innovate are enhanced. Technological advances and improvements in production efficiency can improve resource utilization and reduce energy consumption per unit product, helping firms achieve green manufacturing.

The proportion of patents in manufacturing sales revenue is selected in this study as the proxy variable for technological innovation (Inno) to verify this mechanism. Table 4 reports the corresponding empirical regression results. The regression coefficients for the

technological innovation variable in Column (4) are significantly positive, indicating that technological innovation promotes green competitiveness in manufacturing. Meanwhile, the significantly positive coefficients for the industrial policy variable in Columns (5) and (6) demonstrate that industrial policies have driven technological innovation in the manufacturing industry.

Finally, environmental governance channels are examined. Aside from the introduction of incentive policies by the Chinese government for promoting the growth of encouraging industries, environmental governance measures are not uncommon, including mandatory administrative orders, inspections, direct shutdowns, and relocation of polluting firms. Several scholars have attempted to analyze the functional mechanism of environmental regulation on green transition development in the manufacturing industry. For instance, Matsuhashi et al. [35] claim that environmental regulation produces a significant heterogeneous effect on the green innovation efficiency of the manufacturing industry, where a distinct inverted "U" relationship is present in the high-carbon industry. Dechezleprêtre et al. [36] find a positive correlation between the intensity of environmental governance and the green innovation level of the manufacturing industry. However, the latter is affected by factors such as foreign investment and regional factor endowments. According to the examination of the nonlinear relationship between environmental regulation and the efficiency of the manufacturing industry by Clò et al. [37], the technological and structural effects mainly characterize environmental regulation. When constrained by relatively limited resources, the restrictive pollution control feature of environmental regulation promotes the relative price fluctuations of resource elements. Under stringent supervision and corresponding policy incentives, firms continue to heighten their energy conservation and emissions reduction efforts. By pollution classification and reallocation of production factors, they find more "clean" alternative elements to elevate the environmental standards of production, avoid the loss of economic benefits, and reduce their dependence on traditional resource elements in a "bringing-order-out-of-chaos" method. Consequently, they depend on high-end human capital elements, promoting the high-quality upgrading of element structures and improving allocation efficiency. The proportion of environmental governance investment in manufacturing sales revenue is chosen as the proxy variable of environmental governance (Gov) to verify this mechanism. Table 4 reports the corresponding empirical regression results. The regression coefficients for the environmental governance variable in Column (7) are significantly positive, indicating that environmental governance has enhanced green competitiveness in manufacturing. Meanwhile, the significantly positive coefficients for the industrial policy variable in Columns (8) and (9) demonstrate that industrial policies have driven environmental governance efforts.

## Regulatory role of institutional environment in the industrial policy effects

**Synergy between industrial policy power and market forces.**   China's reform and opening-up follow a development path of "stabilizing stocks and enlarging increments." That is, while driving state-owned enterprises' reform, it promotes the development of foreign-invested firms by "opening to the outside world" and the development of private businesses by "opening to the domestic market." Such institutional innovation has been immensely successful with establishing a market structure in which enterprises of different ownerships compete at the same stage, such as state-owned enterprises, private businesses, and foreign-invested firms. Based on its national conditions, China adopted a pragmatic and progressive course of reform. Specifically, during the early stage of reform, only policy adjustments in individual sectors (e.g., land contracting) were involved. At the Third Plenary Session of the 12th CPC Central Committee, the "socialist commodity economy" goal was proposed. At the 14th CPC

National Congress, the goal of reform was further clarified as the "socialist market economy." During the 15th CPC National Congress, the non-public economy was declared an important constituent of the socialist market economy. The rigid view of linking the proportion of state-owned economy with the nature of socialism was rejected. Eventually, at the Third Plenary Session of the 18th CPC Central Committee, the "fundamental role" of the market in resource allocation was changed to a "decisive role," and the goal of modernizing national governance was proposed. China's practice and understanding of the role of the market economy have gradually deepened. As a major country in reforming toward a socialist market economy, China has an economic, institutional environment and a market participant structure that differs from the Western world. What are the effects of market-oriented reforms on industrial policy implementation in the context of China? Based on the empirical model (1) formula, the multiplication term Policy×Market is added as an explanatory variable, reflecting the policy power and market forces. Thus, the formula is extended to empirical model (2) as follows:

$$lnGTFP_{it} = C + \beta_1 Policy_{it} + \beta_2 Policy_{it} \times Market_{it} + \beta_3 Export_{it} + \beta_4 Capital_{it} + \beta_5 FDI_{it} + V_i + \varepsilon_{it} \quad (2)$$

Columns (1) and (2) of Table 5, the regression results after adding the multiplication term between the policy and market variables are displayed. Regardless of whether the industrial

**Table 5. Moderating effect of institutional environment.**

|  | (1) | (2) | (3) | (4) |
|---|---|---|---|---|
|  | GTFP | GTFP | GTFP | GTFP |
| Policy_acc | 0.095*** |  | 0.200*** |  |
|  | (3.51) |  | (6.09) |  |
| Policy_add |  | -0.016 |  | 0.116*** |
|  |  | (-0.57) |  | (3.35) |
| Policy_acc×FD |  |  | -0.008** |  |
|  |  |  | (-2.12) |  |
| Policy_acc×FD |  |  |  | -0.011** |
|  |  |  |  | (-2.12) |
| Policy_acc×Market | 0.020*** |  |  |  |
|  | (5.57) |  |  |  |
| Policy_add×Market |  | 0.024*** |  |  |
|  |  | (4.07) |  |  |
| Capital | 0.049 | 0.160*** | 0.144*** | 0.234*** |
|  | (1.35) | (4.61) | (3.16) | (5.64) |
| lnFDI | 0.129** | 0.255*** | 0.138** | 0.261*** |
|  | (2.42) | (4.90) | (2.52) | (4.93) |
| Export | 0.158*** | 0.118** | 0.119** | 0.083 |
|  | (3.42) | (2.43) | (2.40) | (1.64) |
| _cons | -1.073** | -1.334** | -1.854*** | -1.930*** |
|  | (-2.13) | (-2.50) | (-3.34) | (-3.42) |
| R$^2$ | 0.566 | 0.513 | 0.534 | 0.496 |
| N | 390 | 390 | 390 | 390 |

*t* statistics in parentheses

* $p < 0.1$

** $p < 0.05$

*** $p < 0.01$.

policies are measured with Policy-acc or Policy_add, the multiplication term Policy×Market is always positive, reaching a 1% significance level. This suggests that the joint action of industrial policy and market forces has prominently promoted the enhancement of GTFP in the regional manufacturing industry.

The marketization heterogeneity of industrial policy effects has profound practical significance and policy implications. The market contributes decisively to resource allocation, and the promotion of industrial policies is conditionally dependent, such as reliance on channels where market forces are at work. This indicates that only those industrial policies that respect and rely on the market mechanism can effectively drive green competitiveness in the manufacturing industry. However, the introduction of industrial policies, disregarding market rules, cannot yield good results. Moreover, in developing countries and regions, industrial policies constitute an effective supplement to the market. Notably, China is a nation with wide regional disparities, where the degree of marketization tends to be lower in backward areas. Although these areas need support from industrial policies, based on the empirical results, the policy effects may not be as good as those with high marketization levels. This requires great attention from policymakers and implementers.

**Impact of fiscal decentralization on the industrial policy effects.** The high economic growth of over 30 years in China is inseparable from its fiscal decentralization reforms initiated in 1979. Qian & Roland [38] propose the theory of "Chinese-style federalism," holding that the administrative system of political centralization and moderate economic decentralization is the fundamental institutional reason for developing the economy by Chinese local governments. Most scholars believe that while the Chinese-style fiscal decentralization has promoted local economic growth in China, it has also become the institutional root for extensive economic development. Under the assessment mechanism in a decentralized context, local officials prefer energy-intensive high-pollution industries with quick returns and high outputs. For short-term economic growth, the local responsible persons have made excessive investments in the infrastructure and other productive fields. Therefore, the low efficiency of resource utilization has hindered the coordinated development of the economy and environment, thereby suppressing the growth of GTFP. Consistent with the view of environmental federalism, the "Race to the Bottom" phenomenon occurs in an environment that is regulated under fiscal decentralization, leading to deteriorated environmental quality [39,40]. Hence, fiscal decentralization is likely to lower the local governments' standards for industrial environmental control, thus becoming a major institutional factor affecting the implementation of industrial policies. In our opinion, an increase in the degree of fiscal decentralization compromises the local governments' efforts to implement industrial policies, thereby weakening policy implementation.

Based on the empirical model (1) formula, the multiplication term Policy×FD is added as an explanatory variable, reflecting policy power and fiscal decentralization. Thus, the formula is extended to empirical model (3) as follows:

$$lnGTFP_{it} = C + \beta_1 Policy_{it} + \beta_2 \text{policy} it \times \text{FDI} it + \beta_3 \text{Export}_{it} + \beta_4 Capital_{it} + \beta_5 FDI_{it} + V_i + \varepsilon_{it} \tag{3}$$

Columns (3) and (4) of Table 5 present the regression results after adding the multiplication term. Regardless of whether the industrial policies are measured with Policy_acc or Policy_add, the multiplication term Policy×FD is always negative, which reaches a 1% significance level. This suggests the weakened role of industrial policies in promoting the green competitiveness of the manufacturing industry with an increasing degree of fiscal decentralization.

## Discussions and policy suggestions

The controversy over the effectiveness of industrial policy runs throughout the development course of economics. The Chinese economy has changed from rapid growth to high-quality development, and China is the world's largest source of carbon emissions. Increasing green total-factor productivity is the key to achieving a change in economic development quality, efficiency, and impetus. Can industrial policy improve the green competitiveness of the manufacturing industry? The attainment of this goal is inseparable from a systematic understanding of the relationship between industrial policy and green TFP of the manufacturing industry, depending on a scientific evaluation of China's institutional environment. Nonetheless, the current literature has some deficiencies in the research on the above issues. First, most studies merely focus on the relationship between industrial policy and labor productivity but neglect the impact of industrial policy on green total-factor productivity. Second, the success of the industrial policy depends on the combined action of elements and institutions, and the institutional environment for successful industrial policy is seldom analyzed. Lastly, when measuring the implementation effect of industrial policy, scholars tend to select an industrial policy indicator that is difficult to depict the whole picture of industrial policy. However, the indicator tends to have an endogenous problem.

In this study, industry-related regional laws and local governmental regulations are collected and collated systematically to build a provincial-level dataset of local industrial regulations for China, used as a proxy indicator for the intensity of local governments' industrial policies. Accordingly, provincial and regional panel data are integrated to empirically examine the driving role of industrial policies in the green competitiveness of the manufacturing industry. Furthermore, the institutional conditions under which industrial policies play an active role are discussed, focusing on investigating the complementary effect between policy power and market forces and the role of fiscal decentralization in the influencing mechanism of industrial policies. It is found that the normative documents on the manufacturing industry are good proxy indicators for the intensity of local governments' industrial policies. The manufacturing regulations from China's provincial-level local governments have helped improve the GTFP of the regional manufacturing industry, suggesting the rationality of governmental intervention. As the mechanism research reveals, resource allocation, innovation incentives, and environmental governance have promoted green competitiveness in manufacturing supported by industrial policies. Additionally, synergy and complementarity exist between governmental policy power and market forces in promoting green competitiveness in manufacturing. The impact of governmental industrial policies on the green competitiveness of the manufacturing industry is based on marketization. The exertion of industrial policies' positive role is closely linked to China's unique fiscal decentralization system. The degree of fiscal decentralization negatively impacts local governments' efforts to implement industrial policies. An increase in the fiscal decentralization degree is detrimental to achieving the expected objectives of industrial policies.

This study's findings have profound policy implications. First, industrial policies can prominently promote the enhancement of GTFP, affirming the existential value of these policies. The "growth discriminating" and "situation exploiting" roles of industrial policies should be grasped scientifically. The industrial policy system should be perfected constantly to promote the green development of the manufacturing industry. Second, the relationship between industrial policies and marketization is symbiotic and complementary, rather than a trade-off relationship. The industrial development strategy should transcend the narrow argument of "market or government," which should be compatible and inclusive. The relationship between policy and the market should be rationalized. Besides giving play to the basic role of the market

in allocating resources to emphasize the "efficient market," it is necessary to exert the government's role in regulating industrial policies. Third, given the conclusion that the current fiscal decentralization system has led to an insufficient supply of local governments' public environmental services, it is imperative to redefine and clarify the environmental governance authority of governments at all levels. New types of relationships between central and local governments that adapt to "high-quality development" should be explored, the incentive distortions of local governments in implementing industrial policies should be corrected as far as possible, and the assessment of local government's implementation of industrial policy objectives should be emphasized, such as the focused assessment of regional shutdown of outdated production facilities and promotion of industrial upgrading.

## Author Contributions

**Conceptualization:** Qing Zhao.

**Data curation:** Qing Zhao.

**Funding acquisition:** Qing Zhao.

**Investigation:** Qing Zhao.

**Methodology:** Qing Zhao.

**Resources:** Chih-Hung Yuan.

**Software:** Qing Zhao.

**Supervision:** Chih-Hung Yuan.

**Writing – original draft:** Qing Zhao.

**Writing – review & editing:** Chih-Hung Yuan.

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
