## [Editor Report · Decision Letter 0]

23 Dec 2020

PONE-D-20-39609

Can China's industrial policies enhance the green competitiveness of manufacturing industry?

PLOS ONE

Dear Dr. Zhao,

Thank you for submitting your manuscript to PLOS ONE. After careful consideration, we feel that it has merit but does not fully meet PLOS ONE’s publication criteria as it currently stands. Therefore, we invite you to submit a revised version of the manuscript that addresses the points raised during the review process.

We look forward to receiving your revised manuscript.

Kind regards,

Ming Zhang, Ph.D.

Academic Editor

PLOS ONE

Journal Requirements:

2.We note that you have indicated that data from this study are available upon request. PLOS only allows data to be available upon request if there are legal or ethical restrictions on sharing data publicly. For more information on unacceptable data access restrictions, please see http://journals.plos.org/plosone/s/data-availability#loc-unacceptable-data-access-restrictions.

Additional Editor Comments (if provided):

You should reorgazied this paper. As you can see, there exists chinese word in the text. More latest literatures should be cited in this paper. Furthermore, five reviewers are needed.
---

## [Author Response · Author response to Decision Letter 0]

12 Jan 2021

Reviewer1：Tze-Yu Wang, bow6974@hotmail.com, National Sun Yat-sen University

Reviewer2：Yu-Hsi Yuan, yuanyh@gm.ypu.edu.tw, Chinese Culture University

Reviewer3：Kebin Deng，ecdengkb@scut.edu.cn，South China University of Technology

Reviewer4：Haijian Zeng, zenghj06@126.com, Guangxi University

Reviewer5：Yi Zhang, zhangyisg@126.com, Guangdong University of Finance

---

## [Decision Letter · Decision Letter 1]

31 Mar 2021

PONE-D-20-39609R1

Can China's industrial policies enhance the green competitiveness of manufacturing industry?

PLOS ONE

Dear Dr. Zhao,

Thank you for submitting your manuscript to PLOS ONE. After careful consideration, we feel that it has merit but does not fully meet PLOS ONE’s publication criteria as it currently stands. Therefore, we invite you to submit a revised version of the manuscript that addresses the points raised during the review process.

We look forward to receiving your revised manuscript.

Kind regards,

Ming Zhang, Ph.D.

Academic Editor

PLOS ONE

Reviewers' comments:

Reviewer's Responses to Questions

**Comments to the Author**

1. If the authors have adequately addressed your comments raised in a previous round of review and you feel that this manuscript is now acceptable for publication, you may indicate that here to bypass the “Comments to the Author” section, enter your conflict of interest statement in the “Confidential to Editor” section, and submit your "Accept" recommendation.

Reviewer #1: All comments have been addressed

Reviewer #2: (No Response)

2. Is the manuscript technically sound, and do the data support the conclusions?

Reviewer #1: Yes

Reviewer #2: Partly

3. Has the statistical analysis been performed appropriately and rigorously? 

Reviewer #1: Yes

Reviewer #2: No

4. Have the authors made all data underlying the findings in their manuscript fully available?

Reviewer #1: Yes

Reviewer #2: No

5. Is the manuscript presented in an intelligible fashion and written in standard English?

Reviewer #1: Yes

Reviewer #2: No

6. Review Comments to the Author

Reviewer #1: This is an interesting and innovative article. Based on the provincial and district level panel data of 30 provinces in China, the industrial policy was quantitatively identified by using local government environmental regulations. To minimize the error of omitted variables, export investment, foreign direct investment and human capital were selected as the control variables. The industrial policy variable, as the core explanatory variable, was defined as two parts: Policy_acc (cumulative number of local regulations and documents on manufacturing industry) and Policy_add (number of newly added local governmental regulations and documents on manufacturing industry). According to established empirical model, the influencing degree and functional mechanism of China’s industrial policies are studied, and the synergistic complementary effect and the role of fiscal decentralization are analyzed. The statistical analysis has been performed appropriately and rigorously. The robustness of the results has also been tested. Therefore, I think this article is worth publishing.

Reviewer #2: In this study, combined with the provincial and regional panel data, the influencing degree and functional mechanism of China's industrial policies on the green competitiveness of manufacturing industry are examined empirically. Respecting functional mechanisms, supported by industrial policies, the resource allocation, innovation incentives and environmental governance have played a positive role in promoting the green competitiveness of manufacturing industry. In addition, the synergistic complementary effect between industrial policy power and market force are investigated, as well as the role of fiscal decentralization in the influencing mechanism of industrial policies.

7. PLOS authors have the option to publish the peer review history of their article (what does this mean?). If published, this will include your full peer review and any attached files.

Reviewer #1: No

Reviewer #2: No

---

## [Author Response · Author response to Decision Letter 1]

14 May 2021

1) and 2): Thank reviewers for offering suggestion on language errors. Due to the deficient language ability, the authors have invited foreign professionals to polish the language of the full text. In the future, we will further improve the language skills to reduce stupid mistakes.

3) Based on the suggestions of reviewers, the author has summed up three contributions of this paper. Firstly, the industrial policy is studied from the perspective of green competitiveness, expanding related research on the implementation effect of industrial policy. Secondly, this paper not only explores the effect of local industrial policies on the green total-factor productivity of the manufacturing industry and specific mechanism, but also enriches the related research on the influencing factors of the green total-factor productivity. Thirdly, the author measures industrial policy from the angle of government regulation and document, verifies the rationality and feasibility of this approach, and expands the measurement perspectives of existing industrial policy research.

4) The author unifies the table format of the whole paper and keeps three decimal points for all the figures in the table.

5) The author supplements the data sources of variables Subs, Inno and Gov (see modification in line 249-277), and makes descriptive statistics for the newly added variables in Table 1.

6) Thank the reviewers for careful correction. Indeed, the control variables of three transmission paths (Subs, Inno and Gov) of industrial policies are different, and putting the control variables in the form will make the regression result disordered. Therefore, the author does not place the control variables in the form, but adds new demonstrations in table 4. Moreover, all regression results contain control variable.

7) The author supplements the data sources of the variables Market and FD (see relevant modification in line 249-277), and makes descriptive statistics of the new variables in Table 1.

---

## [Decision Letter · Decision Letter 2]

14 Jun 2021

Can China's industrial policies enhance the green competitiveness of  the manufacturing industry?

PONE-D-20-39609R2

Dear Dr. Zhao,

We’re pleased to inform you that your manuscript has been judged scientifically suitable for publication and will be formally accepted for publication once it meets all outstanding technical requirements.

Kind regards,

Ming Zhang, Ph.D.

Academic Editor

PLOS ONE

Additional Editor Comments (optional):

Reviewers' comments:

Reviewer's Responses to Questions

**Comments to the Author**

1. If the authors have adequately addressed your comments raised in a previous round of review and you feel that this manuscript is now acceptable for publication, you may indicate that here to bypass the “Comments to the Author” section, enter your conflict of interest statement in the “Confidential to Editor” section, and submit your "Accept" recommendation.

Reviewer #2: All comments have been addressed

2. Is the manuscript technically sound, and do the data support the conclusions?

Reviewer #2: Yes

3. Has the statistical analysis been performed appropriately and rigorously? 

Reviewer #2: Yes

4. Have the authors made all data underlying the findings in their manuscript fully available?

Reviewer #2: Yes

5. Is the manuscript presented in an intelligible fashion and written in standard English?

Reviewer #2: Yes

6. Review Comments to the Author

Reviewer #2: (No Response)

7. PLOS authors have the option to publish the peer review history of their article (what does this mean?). If published, this will include your full peer review and any attached files.

Reviewer #2: No

---

## [Editor Report · Acceptance letter]

21 Jun 2021

PONE-D-20-39609R2 

Can China's industrial policies enhance the green competitiveness of the manufacturing industry? 

Dear Dr. Zhao:

I'm pleased to inform you that your manuscript has been deemed suitable for publication in PLOS ONE. Congratulations! Your manuscript is now with our production department. 

Kind regards, 

on behalf of

Dr. Ming Zhang 

Academic Editor

PLOS ONE